# Security Analysis of Deep Neural Networks Operating in the Presence of Cache Side-Channel Attacks

## Abstract

Recent work has introduced attacks that extract the architecture information of deep neural networks (DNN), as this knowledge enhances an adversary's capability to conduct attacks on *black-box* networks. This paper presents the first in-depth security analysis of DNN fingerprinting attacks that exploit cache side-channels. First, we define the threat model for these attacks: our adversary does not need the ability to query the victim model; instead, she runs a co-located process on the host machine where the victim's deep learning (DL) system is running and passively monitors the accesses of the target functions in the shared framework. Second, we introduce DeepRecon, an attack that reconstructs the architecture of the victim network using the internal information extracted via Flush+Reload, a cache side-channel technique. Once the attacker observes function invocations that map directly to architecture attributes of the victim network, the attacker can reconstruct the victim's entire network architecture. In our evaluation, we demonstrate that an attacker can accurately reconstruct two complex networks (VGG19 and ResNet50) having observed only one forward propagation. Based on the extracted architecture attributes, we also demonstrate that an attacker can build a meta-model that accurately fingerprints the architecture and family of the pre-trained model in a transfer learning setting. From this meta-model, we evaluate the importance of the observed attributes in the fingerprinting process. Third, we propose and evaluate new framework-level defense techniques that obfuscate our attacker's observations. Our empirical security analysis represents a step toward understanding DNNs' vulnerability to cache side-channel attacks.

## 1 Introduction

Deep neural networks (DNNs) have become an essential tool in various applications, such as face recognition, speech recognition, malware detection, and autonomous driving or aviation (Parkhi et al., 2015; Amodei et al., 2016; Arp et al., 2014; Chen et al., 2015; Smolyanskiy et al., 2017). A DNN's performance depends widely on the network architecture—the number and types of layers, how the layers are connected, and the activation functions—and, unfortunately, there is no universal architecture that performs well on all tasks. Consequently, researchers and practitioners have devoted substantial efforts to design various DNN architectures to provide high performance for different learning tasks.

Owing to their critical role, DNN architectures represent attractive targets for adversaries who aim to mount *DNN fingerprinting attacks*. In such an attack, the adversary probes a DNN model, considered confidential, until she infers enough attributes of the network to distinguish it among other candidate architectures. In addition to revealing valuable and secret information to the adversary, DNN fingerprinting can enable further *attacks on black-box models*. While the prior work on adversarial machine learning often assumes a white-box setting, where the adversary knows the DNN model under attack, these attacks are usually unrealistic in practice (Suciu et al., 2018). In consequence, researchers have started focusing on a black-box setting, where model architecture is unknown to the adversary. However, in this setting, the adversary often makes some assumptions about the victim model in order to craft successful adversarial examples (Papernot et al., 2017). Instead of approxi-

mating, the adversary can start by conducting a DNN fingerprinting attack to infer the information required about the model, then use this information to craft adversarial examples that can evade the model. This can also enable model extraction attacks (Tramèr et al., 2016; Kurakin et al., 2016; Wang & Gong, 2018) and membership inference or model inversion attacks (Shokri et al., 2017; Long et al., 2018).

Because of the large number and types of architectural attributes, and the subtle effect that each attribute has on the model's inferences, DNN fingerprinting is challenging when using the typical methods employed in the adversarial machine learning literature. For example, Wang & Gong (2018) propose a hyperparameter stealing attack that requires knowledge of the training dataset, the ML algorithm, and the learned model parameters, yet is unable to extract the model architecture. Wang et al. (2018) demonstrate a fingerprinting attack against transfer learning; however, they rely on the assumption that the teacher model and learning parameters are known to the attacker. To overcome these challenges, recent work has started to investigate attacks that utilize information leaked by *architectural side-channels* on the hardware where the DNN model runs. Hua et al. (2018) extract the network architecture of a model running on a hardware accelerator by monitoring off-chip memory addresses. Yan et al. (2018) reduce the search space from $10^{35}$ to 16 candidates within a given network architecture by exploiting cache side-channels.

In this paper, we ask the question: *how vulnerable are DNNs to side-channel attacks, and what information do adversaries need for architecture fingerprinting?* We perform, to the best of our knowledge, the first *security analysis* of DNNs operating in the presence of cache side-channel attacks. Specifically, we define the *threat model* for these attacks, including the adversary's capabilities and limitations. We then introduce DeepRecon, an efficient attack that reconstructs a black-box DNN architecture by exploiting the Flush+Reload (Yarom & Falkner, 2014) technique, and we further evaluate the importance of specific architectural attributes in the success of fingerprinting. Finally, we propose and evaluate new framework-level defenses against these attacks.

Our attack works by targeting lines of code corresponding to the execution of specific network architecture attributes of a deep learning (DL) framework. Specifically, these lines of code correspond to instructions to execute functions that are mapped into the instruction cache when the functions are invoked. Once these lines of code are identified, our attack flushes them from the instruction cache shared by the attacker and the victim. The attacker waits for the victim's process to run and then measures the time it takes to re-access those same lines of code. If the victim's DNN model has accessed any of these particular functions, the corresponding lines of code will be present in the instruction cache when the attacker tries to re-access them. Therefore, the access time to call these functions will be measurably faster than if the victim had not loaded them back into the shared instruction cache. On the other hand, if the victim DNN model did not access these particular functions, the corresponding lines will not be present in the cache when accessed by the attacker, and thus the access time will be measurably slower. We show that from this seemingly small amount of information that is leaked to the attacker, much of the victim's DNN architecture can be extracted *with no query access required*. To launch this attack, we only assume that: 1) an attacker and a victim are co-located in the same machine, and 2) they use the same shared DL framework.

In evaluations, we demonstrate that, by learning whether or not specific functions were invoked during inference, we can extract 8 architecture attributes across 13 neural network architectures with high accuracy. Based on the extracted attributes, we demonstrate how an attacker can reconstruct the architectures of two common networks, VGG16 (Simonyan & Zisserman, 2014) and ResNet50 (He et al., 2016) as proof of concept. We also demonstrate a useful example of DeepRecon through model fingerprinting in a transfer learning attack. Finally, we propose countermeasures to obfuscate an attacker from extracting the correct attributes and sequences using observation attacks like DeepRecon and show that these defenses significantly increase the errors in the extracted attributes and can be implemented in various DL frameworks without hardware or operating system support.

## 2 BACKGROUND

As opposed to attacks that exploit vulnerabilities in software or algorithm implementations, side-channel attacks utilize information leaks from vulnerabilities in the implementation of computer systems. Due to modern micro-processor architecture that shares the last-level cache (L3 cache) between CPU cores, cache side-channel attacks have become more readily available to implement.

Since the cache is involved in almost all the memory access activities on a machine, it can be a medium that includes abundant information about programs running on the host. The fundamental idea of the attack is to monitor the access time to the shared contents, e.g., shared libraries or credentials, between a victim and an attacker while the attacker fills the cache set with the addresses known to her (Prime+Probe (Liu et al., 2015)) or keeps flushing the shared data from the cache (Flush+Reload (Yarom & Falkner, 2014)). In both the cases, once the victim accesses memory or shared data, the attacker can identify which memory addresses or shared data is accessed. Prior work has demonstrated that, with cache side-channels, an attacker can construct covert channels between processes, stealing cryptographic keys, or breaking the isolation between virtual machines (Zhang et al., 2014; Liu et al., 2015).

### FLUSH+RELOAD

Our attack leverages the Flush+Reload technique, which monitors accesses to memory addresses in shared contents. The technique assumes that an attacker can run a spy process on the same host machine. This enables the attacker to monitor the shared data or libraries between her and the victim. During monitoring, the attacker repeatedly calls the `clflush` assembly instruction to evict the L3 cache lines storing shared content and continually measures the time to reload the content. A fast reload time indicates the data was loaded into the cache by the victim whereas a slow reload time means the data is not used. From this information, the attacker determines what data is currently in use and identifies the control flow (order of function calls) of the victim's process. We chose Flush+Reload over Prime+Probe since the results from Flush+Reload produce less noise.

### ATTACKS ON BLACK-BOX DEEP NEURAL NETWORKS

Prior work has proposed various methods to attack black-box DNNs. Tramèr et al. (2016) and Papernot et al. (2017) demonstrated model extraction attacks on black-box DNNs that aim to learn a substitute model by using the data available to the attacker and observing the query results. Fredrikson et al. (2015) and Shokri et al. (2017) demonstrated model inversion attacks on black-box DNNs that reveal a user's private information in the training data leveraging model predictions. Wang & Gong (2018) proposed a hyper-parameter stealing attack that aims to estimate the hyper-parameter values used to train a victim model. However, these attacks require unrealistic conditions, e.g., the architecture of the victim network needs to be known to attackers, or the victim uses a network with simple structures, such as multi-layer perceptrons. Thus, the capability of DeepRecon attack that reconstructs black-box DNN architectures can bridge the gap between the realistic black-box scenario and their conditions.

### RECONSTRUCTING BLACK-BOX DNNS VIA SIDE-CHANNELS

Recent studies have discovered various methods to extract the architecture of a black-box DNN.

**Memory and Timing Side-Channels:** Hua et al. (2018) monitored *off-chip memory accesses* to extract the network architecture of a victim model running on a hardware accelerator. They estimated the possible architecture configurations and extracted model parameters. However, the attack requires physical accesses to the hardware, whereas our attack does not.

**Power Side-Channel:** Wei et al. (2018) demonstrated that an attacker can recover an input image from collected power traces without knowing the detailed parameters in the victim network. However, this approach also assumed an attacker who knows the architecture of a victim network, so our attack could help meet the assumptions of this attack as well.

**Cache Side-Channel:** Concurrent work by Yan et al. (2018) demonstrates that an attacker can reveal the architecture details by reverse engineering and attacking generalized matrix multiply (GeMM) libraries. However, GeMM-based reverse engineering can only reveal the number of parameters of convolutional or fully connected layers because others such as activation and pooling layers are difficult to characterize by matrix multiplications. Also, in order for the monitored functions in GeMM libraries to be in a shared instruction cache of an attacker and a victim, the multiplications must occur on the CPU. However, DeepRecon can be performed independent of the hardware on which the computations occur, generalizing better common hardware on which DNNs run (e.g., GPUs).

**Using Known Student Models:** Wang et al. (2018) proposed a transfer learning technique in which an attacker identifies teacher models by using known student models available from the Internet. This approach assumed that the victim selects the teacher from a set of known architectures. We, however, take this a step further and fingerprint families of architectures as well as many commonly known teacher models. Additionally, we are able to reconstruct arbitrary teacher model architectures with high accuracy.

**Meta-Models:** Oh et al. (2018) demonstrated that an attacker can estimate the victim's architecture by using a brute-force approach and meta-models. They first trained all the possible architectures of a given set and pruned the models with inferior performance. Then, they trained a meta-model that identified the network architecture using mutated samples and labels. However, the pruning process is time intensive (i.e., 40 GPU days for 10k candidates of LeNet (LeCun, 1998)), and the candidates were selected from limited architectural choices, whereas we again go a step further in identifying families of architectures and can generalize to previously unknown teacher models.

## 3 DeepRecon Attack

### 3.1 Threat Model

Our threat model requires an attacker who can launch a *co-located user-level process* on the same host machine as the victim. This ensures the attacker and the victim's process share the same instruction cache. This co-location also allows our attacker to observe the victim DNN's behavior without actively querying the model, avoiding the common assumption of query access in the literature on black-box attacks. Consider the example of any computer that an attacker has access to at a user-level, the attacker can log into this machine and attack other users with DeepRecon. Another way for an attacker to achieve co-location is to disguise her process as a benign program such as an extension for a browser. Once some victims install the extension in their browser, the attacker can easily launch a monitoring process. We also assume that the attacker and victim use *the same open-source DL frameworks* shared across users. Importantly, this assumption can be easily met because many popular DL frameworks such as Tensorflow (Abadi et al., 2016) or PyTorch[1] are provided as open-source libraries, and this practice of sharing libraries across users is default on major operating systems, e.g., Windows, MacOS, and Ubuntu. Thus, our attacker can identify the addresses of functions to monitor in the instruction cache by reverse-engineering the shared framework's code.

**Motivating Attack Example:** We provide a practical example where our threat model is applicable. Suppose an attacker aims to install malware on a victim's machine where an anti-virus system, based on a DNN model, is running. To evade malware detection in common black-box attacks such as the attack proposed in Ilyas et al. (2018), an attacker needs to drop crafted programs actively to monitor the model's decisions and synthesize an evasive sample based on the collected data. However, when the attacker drops multiple files, her behavior can be detected by the victim. This is further amplified by the need to query the model repeatedly to craft any more malicious files.

On the other hand, our attacker induces the victim to install a chrome add-on (which runs at a user-level) that passively monitors cache behaviors of the model and extracts the architecture. Then, the attacker trains a surrogate model with public datasets (including malware and benign software). With the surrogate model, the attacker crafts her malware that evades detection and can continue to craft malicious files that will be classified as benign offline and without any further observations. As opposed to common black box attacks, our attacker lowers the possibility of being caught because she only monitors the victim model while it is in use and does not need to query the model.

### 3.2 Attack Overview

The overview of DeepRecon attack is described in Fig. 1. The victim's behaviors are depicted with the dotted lines (black), and the attacker's actions are described with the solid lines (red). While preparing the attack offline, the attacker first analyzes the deep learning framework that the victim uses and collects the target functions corresponding to the architecture attributes that the attacker wants (Table 1). Then later, the attacker launches a co-located process at the user-level that runs along with the victim's process on the same host machine. When the victim's process runs training

---

[1]https://pytorch.org

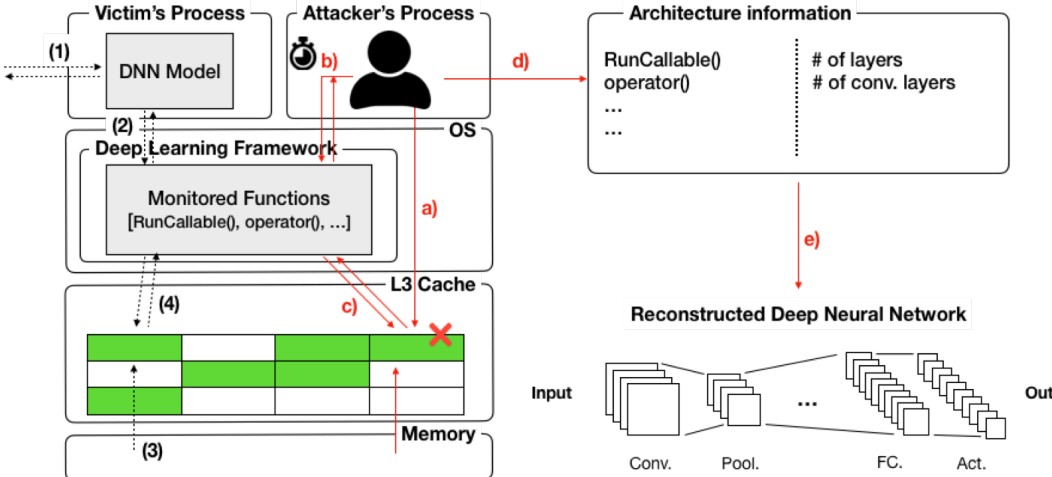

Figure 1: **DeepRecon Visualization.** During forward/backward propagation, the victim DNN model computes the prediction using the functions in the DL framework [(1), (2), (3), (4)]. Since the attacker and victim share an instruction cache, the attacker can invalidate the cache lines that store the functions [a)], and then monitor if the victim calls these functions by measuring the access time [b), c)]. Based on the mapping between the functions and the architecture attributes, the attacker reconstructs the architecture of the victim's DNN [d), e)].

or predictions with its model, the target functions are invoked and the instructions that call them are loaded into the shared instruction cache. The attacker periodically flushes the cache lines and measures the access time to the target instructions. If the victim invokes any of the target functions after flushing, the following access time measured by the attacker will be measurably faster than if the victim does not invoke them. The attacker collects the number and sequence of invocations and then extracts the victim model's architecture attributes. Then, the attacker reconstructs the victim model's architecture.

### 3.3 REVERSE ENGINEERING

In Table 1, we analyze the TensorFlow v1.9.0-rc0 framework[2] and list the target functions corresponding to the architecture attributes. We choose TensorFlow due to its popularity as an open source machine learning (ML) framework, but believe that the methods we describe will be applicable to most, if not all, other popular frameworks. In addition to having found some corresponding functions in another popular framework, PyTorch/Caffe2, our attack leverages the inherent structure of a scalable and widely deployable ML framework, namely library layer abstraction. All the of the functions we monitor in TensorFlow are in the core of the library, both below the API interface and above the system dependent code. Because of this, our attack not only does not depend on the specific TensorFlow API a victim uses but also is agnostic to the type of processing hardware the victim is using, from a single CPU to a cluster of GPUs.

The specific functions we monitor in Table 1 represent two subgroups: those corresponding to control flow and those corresponding to architecture attributes. The control flow functions allow us to observe the number of queries the victim makes to the model and the number of layers that are updated by gradient decent if we observe the model when it is being trained The function that monitors the number of queries is especially important, as it allows us to separate individual observations. The architecture attribute functions are called once per instance of an architecture attribute being present in the neural network, allowing us to see the number of each attribute and the sequence in which they occur in the victim's architecture. Combined, these functions allow us to observe the architecture attributes of a neural network from start to finish on a given observation.

Additionally, the bias operator gradient function, given in the table by #grads, can allow an attacker to figure out the total number of layers that are updated during training time if the attacker observes

---

[2]https://github.com/tensorflow/tensorflow/releases/tag/v1.9.0-rc0

| Type | Code | Stage | Func. Name | Location in TensorFlow Code |
|------|------|-------|------------|------------------------------|
| **Control Flow** | #queries | T/I | RunCallable() | core/common_runtime/session_ref.cc [line: 154] |
| | #grads | T | compute() | core/kernels/bias_op.cc [line: 218] |
| | | | | |
| | #convs | T/I | operator() | core/kernels/conv_ops.cc [line: 122] |
| | #fcs | T/I | compute() | core/kernel/matmul_op.cc [line 451] |
| | #softms | T/I | compute() | core/kernels/cwise_ops_common.h [line: 240] |
| **Arch. Attributes** | #relus | T/I | compute() | core/framework/numeric_op.h [line: 58] |
| | #mpools | T/I | compute() | core/kernels/pooling_ops_common.h [line: 109] |
| | #apools | T/I | compute() | core/kernels/avgpooling_op.cc [line: 76] |
| | #merges | T/I | compute() | core/kernels/cwise_ops_common.h [line: 91] |
| | #biases | T/I | compute() | core/kernels/bias_op.cc [line: 98] |

(Note that **T** stands for the training, and **I** indicates the inference.)

Table 1: **Target Functions.** The monitored functions in the TensorFlow framework (v1.9.0-rc0). Each function corresponds to a control flow or an attribute. [Note that the codes are the number of queries (*#queries*), gradient updates (*#grads*), convolutional layers (*#convs*), fully connected layers (*#fcs*), softmaxs (*#softms*), ReLUs (*#relus*), max poolings (*#mpools*), avg. poolings (*#apools*), merge operations (*#merges*), bias operations (*#biases*).]

the training of the model. Using this information, the attacker, already knowing the total number of layers in the architecture, can find the point at which the victim is freezing the backpropagation. This allows the attacker to know which layers are directly inherited from the training model and which layers are specifically trained by the victim. The relevance of this will be discussed in our application of DeepRecon to model fingerprinting (Sec. 4).

**Limitations.** Similar to concurrent work (Yan et al., 2018), we are also able to extract additional information, such as the number of parameters in convolutional and fully connected layers by monitoring the matrix multiplications in the Eigen library[3] on which TensorFlow is built. This attack provides more fine-grained information, but it does not generalize to computations on hardware other than a CPU. Also, we examine whether our attack can recover the inputs to the model and its parameters. By varying inputs and parameters while monitoring the functions used to compute these parameters using a code coverage tool, GCOV[4], we find that the framework implements matrix multiplications of parameters in a data-independent way. Thus, we are unable to estimate the inputs and parameters of a victim model. We hypothesize that this is a general limit of cache based side-channel attacks on DNNs that target instructions, and that obtaining the parameters is reducible to the problem of reading arbitrary victim memory.

### 3.4 Extracting Architecture Attributes

We run our attack on Ubuntu 16.04 running on a host machine equipped with the i7-4600M processor (8 cores and 4MB L3 cache). Our victim and attacker processes are running at the user-level on the same operating system (OS). Both the processes utilize the TensorFlow v1.9.0-rc0 framework. The victim uses the DNN model to make predictions, and the attacker launches the Flush+Reload attack using the Mastik toolkit (Yarom, 2016) to monitor the target functions at the same time.

A total of 13 convolutional neural network (CNN) architectures are considered in our experiment: DenseNet121, 169, 201 (Huang et al., 2017), VGG16, 19 (Simonyan & Zisserman, 2014), ResNet50, 101, 152 (He et al., 2016), InceptionV3, InceptionResNet (Szegedy et al., 2015), Xception (Chollet, 2017), MobileNetV1, and MobileNetV2[5] (Howard et al., 2017).

Table 2 includes the extraction results from monitoring VGG16 and Resnet50. The full extraction results from the 13 networks are in Appendix D. We first show the results from a **S**hort attack, where an attacker can only run her process on a short interval of time, observing only a single query of the

---

[3]http://eigen.tuxfamily.org/index.php

[4]https://gcc.gnu.org/onlinedocs/gcc/Gcov.html

[5]Note that we use $alpha = 1.0$ for both.

| Arch. | Data | Attributes | | | | | | | | Errors |
|---|---|---|---|---|---|---|---|---|---|---|
| | | #convs | #fcs | #softms | #relus | #mpools | #apools | #merges | #biases | |
| **VGG19** | **G** | 16 | 3 | 1 | 18 | 5 | 0 | 0 | 19 | - |
| | **S** | 16.2 | 3 | 1 | 18 | 5 | 0 | 0.6 | 18.7 | 1.1/62 |
| | **L** | 16.3 | 3 | 1 | 18 | 4.9 | 0.1 | 1.6 | 18.8 | 2.3/62 |
| **ResNet50** | **G** | 53 | 1 | 1 | 49 | 1 | 1 | 16 | 50 | - |
| | **S** | 54.7 | 1 | 1 | 48.9 | 0.9 | 1.1 | 15.9 | 49.8 | 2.3/173 |
| | **L** | 54.5 | 1 | 1 | 48.9 | 1.1 | 1 | 16 | 49.8 | 2.9/173 |

(Note that **G**, **S**, and **L** means **G**round truth, and the observations from **S**hort and **L**ong attacks.)

Table 2: **Observed Architecture Attributes of VGG19 and ResNet50.** We consider the 8 attributes extracted from short and long attacks. In the short attacks, We report the average values observed in ten random queries whereas the long attacks include the averages of ten continuous queries.

network. We randomly choose ten individual queries and average the attributes. We report errors as the sum of absolute deviations from ground truths. In VGG19, our attacker has 2.6 errors on average and 3.1 in ResNet50. We also show the extraction results from 10 continuous observations (**L**), in which the attacker runs her process for a more extended period of time. The error rates in both the networks are similar. These results demonstrate that DeepRecon achieves better accuracy by only observing a running network than prior work that assumes query access (Oh et al., 2018).

## 3.5 RECONSTRUCTING THE ARCHITECTURE OF BLACK-BOX DEEP NEURAL NETWORKS



(a) ConvNet (**VGG**)    (b) Identity (**ResNet**)    (c) Residual (**ResNet**)

Figure 2: **Basic Building Blocks of VGG and ResNet Architectures.** The feature extractor of a CNN is composed of basic building blocks. We describe the blocks used in VGGs and ResNets. (Note that we exclude the normalization layers from each figure.)

Based on the extracted architecture information, DeepRecon reconstructs the entire DNN architecture of the victim model. In these examples, we focus on the fact that most CNN architectures consist of the feature extractor and classifier layers. The feature extractor is located in the earlier layers and is the combination of basic building blocks. The classifier is a set of fully connected layers at the end of the network. In VGGs and ResNets, there are standard blocks used in each CNN architecture as we depict in Fig. 2. Each block includes activation layers with preceding convolutional layers. In the classifier layers, each fully connected layer is followed by an activation layer.

We describe the reconstruction process of ResNet50 in Table 3. (Note that we also reconstructed VGG16 without errors and show the result in Appendix A.) In this table, we compare the computation sequences observed by the attacker with the actual computations in ResNet50. We can see the sequences are accurately captured with a few errors. The three steps at the bottom describe the reconstruction processes of our attacker. Our attacker first identifies *(1) the number of blocks* by counting the (max-)pooling layers. Once the attacker separates blocks with the pooling layer locations, she counts *(2) the number of convolutional layers* in each block. In ResNets, we know that the Residual block has four convolutional layers, and the Identity block has three convolutional layers in each. Thus, the attacker can identify the type of each block. After that, the attacker estimates *(3) the number of fully connected layers* at the end. Finally, with this block-level information, our attacker successfully estimates the victim architecture is the ResNet50 with high accuracy.

**Discussion about the Reconstruction Errors.** We also examine whether the errors in our experiments have specific patterns, allowing our attacker to filter them out. However, we could not find any pattern: the types and locations of error attributes are different in each run (over 10 runs). Thus,

| Arch. | Data | Computations Sequences (Layers in the Ground Truth) | | | | | | |
|---|---|---|---|---|---|---|---|---|
| **ResNet50** | **G** | $C_R\,P_M$ | $C_R\,C_R\,C\,C\,M_R$ | $C_R\,C_R\,C\,M_R$ | $C_R\,C_R\,C\,M_R$ | | | |
| | | | $C_R\,C_R\,C\,C\,M_R$ | $C_R\,C_R\,C\,M_R$ | $C_R\,C_R\,C\,M_R$ | $C_R\,C_R\,C\,M_R$ | | |
| | | | $C_R\,C_R\,C\,C\,M_R$ | $C_R\,C_R\,C\,M_R$ | $C_R\,C_R\,C\,M_R$ | $C_R\,C_R\,C\,M_R$ | $C_R\,C_R\,C\,M_R$ | $C_R\,C_R\,C\,M_R$ |
| | | | $C_R\,C_R\,C\,C\,M_R$ | $C_R\,C_R\,C\,M_R$ | $C_R\,C_R\,C\,M_R$ | $P_A\,F_{So}$ | | |
| | **S** | $C_R\,P_M$ | $\mathbf{C}\,C_R\,\mathbf{C_R}\,C\,M_R$ | $C_R\,C_R\,C\,M_R$ | $C_R\,C_R\,C\,M_R$ | | | |
| | | | $\mathbf{C}\,C_R\,\mathbf{C_R}\,C\,M_R$ | $C_R\,C_R\,\textbf{...}\,M_R$ | $C_R\,C_R\,C\,M_R$ | $C_R\,C_R\,C\,M_R$ | | |
| | | | $\mathbf{C}\,C_R\,\mathbf{C_R}\,C\,M_R$ | $C_R\,C_R\,C\,M_R$ | $C_R\,C_R\,C\,M_R$ | $C_R\,C_R\,C\,M_R$ | $C_R\,C_R\,C\,M_R$ | $\color{red}{\mathbf{R}}\,C_R\,C\,M_R$ |
| | | | $\mathbf{C}\,C_R\,\mathbf{C_R}\,C\,M_R$ | $C_R\,C_R\,C\,M_R$ | $C_R\,C_R\,C\,M_R$ | $P_A\,F_{So}$ | | |

| Recon. | Steps | Details | | | | | | |
|---|---|---|---|---|---|---|---|---|
| **ResNet50 Recon.** | **(1)** | Block 1. | Block 2. | Block 3 | Block 4. | | | |
| | | | Block 5. | Block 6. | Block 7. | Block 8. | | |
| | | | Block 9. | Block 10. | Block 11. | Block 12. | Block 13. | Block 14. |
| | | | Block 15. | Block 16. | Block 17. | Block 18. | | |
| | **(2)** | Input | Residual Block | Identity Block | Identity Block | | | |
| | | | Residual Block | Identity Block | Identity Block | Identity Block | | |
| | | | Residual Block | Identity Block | Identity Block | Identity Block | Identity Block | Identity Block |
| | | | Residual Block | Identity Block | Identity Block | Fully Connecteds | | |
| | **(3)** | ResNet 50: Configuration with 50-Layers | | | | | | |

(Note that $C$,$P$,$F$,$M$ indicate the $C$onvolutional, $P$ooling, $F$ully connected, $M$erge layers, and the subscripts mean the activations ($R$: ReLU and $So$: Softmax).)

Table 3: **Reconstruction Process of ResNet50 Architecture.** We list the computation sequences captured by our attack in the above rows and the reconstruction process at the bottom rows. The errors in capturing the correct computation sequences by our attacker are marked as bold and red.

we attribute these errors to two primary causes. First, there can be background noise from other processes that our Flush+Reload attack picks up, e.g., a process can pull data into the L3 cache and evict the target function between when the victim calls the function and we reload it. In this case, our attacker cannot observe the victim calling the function. Second, our attack can experience common errors associated with the Flush+Reload attack (Yarom & Falkner, 2014), e.g., a victim invokes the target function when we reload, causing our attacker to see a cache miss instead of correctly observing a cache hit.

## 4 FINGERPRINTING BLACK-BOX NEURAL NETWORKS

Our attacker identifies victim neural network architectures using statistical models trained on the attributes as features and the architectures as labels. This is a powerful capability to have if our attacker aims to fingerprint the architecture of the pre-trained models used in transfer learning. Transfer learning is typically done in a fine-tuned manner: a user creates a student model that uses the architecture and parameters of a teacher model and trains only a few fully connected layers at the end of the network by freezing the backpropagation of the preceding layers. We also found that our attacker can learn the layers in the student model not updated during training when they observe the training process (Sec. 3.3). Thus, our attacker can extract both the network architecture and frozen layers of the student (victim) model.

Once our attacker identifies the victim network's teacher model with this information, the attacker can utilize several pre-trained models available from the Internet to perform further attacks. For instance, an attacker can increase the success rate of black-box attacks by crafting adversarial samples with the internal representation from the teacher models (Wang et al., 2018). Additionally, since adversarial samples transfer across different models for the same task (Tramèr et al., 2017), the attacker is not required to use the exact same teacher model—i.e., she can use any pre-trained model of an architecture family that achieves similar accuracy on a task. An attacker can also perform model extractions easily. This is because the model parameters from the pre-trained models can be readily found on the Internet as well and are used in a victim model in this setting. Finally, if the attacker has a partial knowledge of the victim's training data and can gain the knowledge of which layers were frozen during training (see Sec. 3.3), she can fully estimate the parameters of the entire network by independently training the last few layers that were not frozen.

To evaluate our fingerprinting attack, we train decision tree classifiers on the 13 networks used in Sec. 3.4 to identify the network architectures using the extracted attributes and labels. We extract the attributes over 50 observations of each network (650 in total) and utilize 5-fold cross-validations. We

| Task | Networks | Acc. [Avg.] | Important Attributes | | | |
|------|----------|-------------|----------|----------|----------|----------|
| **Total** | - | 1.0 [0.9046] | #relus [0.2575] | #merges [0.2534] | #convs [0.2497] | #biases [0.1034] |
| **Family** | - | 1.0 [0.9938] | #relus [0.4621] | #convs [0.4421] | #mpools [0.3382] | #apools [0.2752] |
| **Arch. Variants** | V | 1.0 [0.9867] | #relus [0.6982] | #convs [0.6982] | #biases [0.6898] | - |
| | R | 1.0 [0.9900] | #relus [0.6399] | #merges [0.6399] | #convs [0.6399] | #biases [0.3750] |
| | D | 1.0 [0.9867] | #relus [0.6399] | #merges [0.6399] | #convs [0.6100] | - |
| | I | 1.0 [1.0000] | #convs [0.6399] | #merges [0.6399] | #apools [0.5875] | #biases [0.3373] |
| | M | 1.0 [1.0000] | #relus [0.6982] | #convs [0.6982] | #fcs [0.6595] | #softms [0.6228] |

(Note that **V**, **R**, **D**, **I**, and **M** indicate **V**GGs, **R**esNets, **D**enseNets, **I**nceptionNets, and **M**obileNets.)

Table 4: **Fingerprinting Performance and Important Attributes.** Each row corresponds to each task. We list the accuracy of the best classifiers and the essential attributes based on the MI scores, denoted by the numbers in brackets.

measure the classification accuracy and analyze the four most essential attributes based on mutual information (MI) scores. Since the attributes are not affected by the host machines or operating systems, the attacker can train the models offline for use in attacks.

Table 4 shows the results of fingerprinting the neural networks. We conduct three types of classification tasks with the aim of identifying 1) the entire 13 networks, 2) 5 network families, and 3) architecture variants in each network[6]. We report the accuracy of best decision trees and the average accuracy over the cross-validations. In all the tasks, our decision trees achieve 100% accuracy, which demonstrates, once trained, these statistical models can be perfect predictors. (Note that we also visualize our data in an attribute space via PCA analysis in Appendix C). We also identified the four essential attributes across all the classifications: 1) *#relus*, 2) *#merges*, 3) *#convs*, and 4) *#apools*. Identifying these influential attributes can guide a potential obfuscation-based defensive strategy against such side-channel attacks.

## 5 DEFENSES TO DEEPRECON ATTACK

Previous studies on defenses against cache side-channel attacks (Kong et al., 2013; Zhou et al., 2016) require specific hardware (page-locked cache) or kernel-level features (page coloring). These solutions have not been widely deployed or have remained as optional features because of their impact on computational performance. Hence, we propose framework-level defenses that do not require specialized hardware or kernel-level updates. Our findings in Sec. 4 regarding the essential architecture attributes for an attack, e.g., *#relus*, *#convs*, and *#merges*, guide our search for defenses. As a result, we propose *obfuscating* the attacker's observations of these attributes. We show that these defenses significantly reduce the success of our DeepRecon attack, and they can be extended to protect against a more general class cache side-channel attacks against DL frameworks.

### 5.1 RUNNING DECOY PROCESSES WITH TINY MODELS

DeepRecon and other potential cache side-channel attacks on deep learning frameworks can only observe that a library function is called, but not by whom. By running an extra process (i.e., a *decoy process*) simultaneously with the actual process, we develop a simple but effective defensive strategy. The decoy process also invokes the target functions in the shared framework, which obfuscates the architecture attributes and computation sequences. To minimize the computational overhead, we utilize networks that only have a few layers, referred to as *TinyNets*. To evaluate our defense, we train these *TinyNets* at the same time as the victim's process is running ResNet50, and we measure the number of errors in the extracted attributes. The results are listed in Table 5. We experiment with three TinyNets: 1) only with a Conv. layer (C:1), 2) Conv. with a ReLU layer (C:1, R:1), and 3) two Conv. and ReLU layers with a Merge layer.

We found the existence of a decoy process significantly hinders the attacker's ability to extract correct attributes, causing 1283-2211 errors in the attribute extractions. Contrasted with up to 2.9 errors on average from DeepRecon's previous extractions (Section 3.4), we find that this defense is

---

[6]In the task 3), we consider MobileNet and MobileNetV2 as the same family.

| Network | Arch. | Attributes | | | | | | | | Errors | Time |
|---------|-------|------------|---|---|---|---|---|---|---|--------|------|
| | | #convs | #fcs | #softms | #relus | #mpools | #apools | #merges | #biases | | |
| **ResNet50** | - | 54.5 | 1 | 1 | 48.9 | 1.1 | 1 | 16 | 49.8 | 2.9 | 17.88 |
| **ResNet50 + TinyNets** | **C:1** | 368.75 | 1.05 | 1.05 | 47.05 | 0.95 | 1.00 | 687.75 | 347.80 | 1282.40 | 23.79 |
| | **C:1 R:1** | 360.00 | 1.15 | 1.15 | 394.00 | 1.00 | 1.00 | 675.95 | 350.80 | 1612.05 | 23.85 |
| | **C:2 R:2 M:1** | 414.55 | 1.00 | 1.00 | 715.10 | 1.10 | 1.00 | 782.25 | 468.25 | 2211.25 | 26.26 |

Table 5: **Effectiveness of the Decoy Process.** We compare the 8 attributes extracted from 10 runs, average errors, and average time with and without TinyNets. Note that **C** refers to the number of convolutional layers, **R** refers to the number of relu activation layers, and **M** refers to the number of merge layers.

exceedingly effective at curbing this type of reconstruction attack. We also show that we can increase the errors associated with the attributes that we aim to obfuscate. For instance, when we run the TinyNet with only one convolutional layer, we observe the #convs is significantly increased. This is important because, with our defenses, a defender can choose the attributes to obfuscate. Since the defender can control what noise gets introduced, they can also dynamically and adaptively change what noise is added into the attackers observations, thereby increasing our defenses effectiveness and generalizability. To quantify the overhead required of this defense, we measure the average network inference time with and without a decoy process, and we observe that the defense increases the inference time by only 5.91-8.38 seconds per inference. Thus, this defense is a reasonable measure to combat cache side-channel attacks that reconstruct DNNs.

## 5.2 Oblivious Model Computations

Another defense against DeepRecon is to obfuscate the order and number of the computations (i.e., function invocations) observed by our attacker using oblivious model computations. We propose two approaches. First, we can update the victim's architecture by adding extra layers. To minimize the side-effects such as performance loss, these layers return the output with the same dimensions as the input. For instance, the convolutional layer with kernel size 3, padding size 1, and strides of length 1 can preserve the input dimensions, and the identity block of the ResNets preserves this as well. Thus, we augment the original architecture by adding such layers at the random location to make the same architecture look different in the attacker's point of view.

Prior work (Targ et al., 2016) has shown the *unraveled view* of the ResNet architecture. Under this view, the skip-connections of a network can be expressed as the ensemble of multiple computational paths that can be computed independently. Hence, we try splitting a computational path with skip-connections into multiple paths without skip-connections. In forward propagation, the multiple paths are randomly chosen and computed so that our attacker finds it difficult to capture the exact architecture. To evaluate our intuition, we construct the obfuscated architecture of ResNet50 (see Appendix B) and extract the attributes by using our attack. Our results are shown in Table 6.

| Arch. | Data | Attributes | | | | | | | | Errors | Time |
|-------|------|------------|---|---|---|---|---|---|---|--------|------|
| | | #convs | #fcs | #relus | #softms | #mpools | #apools | #merges | #biases | | |
| **ResNet50** | **G** | 53 | 1 | 1 | 49 | 1 | 1 | 16 | 50 | - | 17.88 |
| | **S** | 64.80 | 1 | 1 | 57 | 1 | 1 | 20.3 | 54.9 | 29/173 | 24.03 |

(Note that **G**, and **S** means **G**round truth and **S**hort attack.)

Table 6: **Extracted architecture attributes with Obfuscated ResNet50.** We obfuscated the first 3 blocks (i.e., *Residual*, *Identity*, and *Identity* blocks) of the ResNet50 architecture.

Using this defense, the errors detected by DeepRecon increased from 2-3 to 28 for ResNet50. During this test, the first 3 blocks over the entire 16 blocks of ResNet50 are obfuscated. While less effective than our previous defense, we conclude that this defense still can marginally obfuscate the observations of our attacker. Additionally, the gain on computational time is also small: it only increased from 17.88 to 24.03 seconds.

# 6 CONCLUSION

This paper conducts the first in-depth security analysis of DNN fingerprinting attacks that exploit cache side-channels. We first define the realistic threat model for these attacks: our attacker does not require the ability to query the victim model; she runs a co-located process on the machine where the victims DL system is running and passively monitors the accesses of target functions in a shared framework. We also present DeepRecon, an attack that reconstructs the architecture of a victim network using the architecture attributes extracted via the Flush+Reload technique. Based on the extracted attributes, we further demonstrate that an attacker can build a meta-model that precisely fingerprints the architecture and family of the pre-trained model in a transfer learning setting. With the meta-model, we identified the essential attributes for these attacks. Finally, we propose and evaluate new framework-level defense techniques that obfuscate our attackers observations. Our empirical security analysis represents a step toward understanding how DNNs are vulnerable to side-channel attacks.

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

APPENDIX

## A   RECONSTRUCTION OF VGG16 ARCHITECTURE

| Arch. | Data | Computations Sequences (Layers in the Ground Truth) | | | | | |
|-------|------|------|------|------|------|------|------|
| **VGG16** | G | $C_R\,C_R\,P$ | $C_R\,C_R\,P$ | $C_R\,C_R\,C_R\,P$ | $C_R\,C_R\,C_R\,P$ | $C_R\,C_R\,C_R\,P$ | $F_R\,F_R\,F_{So}$ |
| | S | $C_R\,C_R\,P$ | $C_R\,C_R\,P$ | $C_R\,C_R\,C_R\,P$ | $C_R\,C_R\,C_R\,P$ | $C_R\,C_R\,C_R\,P$ | $F_R\,F_R\,F_{So}$ |
| **Recon.** | **Steps** | **Details** | | | | | |
| **VGG16 Recon.** | **(1)** | Block 1. | Block 2. | Block 3. | Block 4. | Block 5. | - |
| | **(2)** | ConvNet (2) | ConvNet (2) | ConvNet (3) | ConvNet (3) | ConvNet (3) | Fully Connected |
| | **(3)** | VGG16: ConvNet Configuration 'C', 'D' | | | | | |

(Note that $C,P,F$ indicate the $C$onvolutional, $P$ooling, $F$ully connected layers, and the subscripts mean the activation functions ($R$: ReLU, and $So$: Softmax).)

Table 7: **Reconstruction Process of VGG16 Architecture.** We list the computation sequences captured by our attack above and the reconstruction process at the bottom.

We describe the reconstruction process of VGG16 in Table 7. The upper table indicates the sequences that our attacker captured, and the bottom shows the actual reconstruction steps. Our attacker identifies the basic building blocks by splitting the sequence with pooling layers. Then, the attacker counts the number of convolutional layers in each block. In VGG16, we found the first two ConvNet blocks have two convolutional layers, and the next three ConvNet blocks have three convolutional layers in each. Additionally, the attacker estimates the number of fully connected layers attached at the end. Once the attacker recovers all the blocks, our attacker can identify the victim architecture as being VGG16 with the ConvNet configuration 'C' or 'D'.

## B   OBFUSCATED RESNET50 ARCHITECTURE

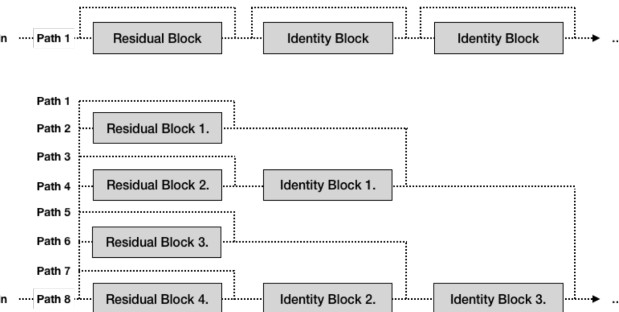

Figure 3: **Unraveled View of the First Three Blocks in ResNet50.** The first three blocks in the original ResNet50 are in the upper diagram, and we unravel them as the architecture at the bottom.

In Sec. 5.2, we construct the obfuscated ResNet50 architecture by using the unraveled view of the first three blocks in ResNet50 as shown in Fig. 3. The upper architecture depicts the block connections in the original ResNet50. In this network, the blocks are computed sequentially, e.g., Residual Block → Identity Block → Identity Block. However, in our unraveled architecture at the bottom, there are individual 8 paths that can be computed independently. We use this architecture to compute the blocks as follows: Residual Block 1, 2 → Identity Block 1 → Residual Block 3, 4 → Identity Block 2 → Identity Block 3. This makes our attacker have difficulty in estimating the architecture attributes and computation sequences of ResNet50.

## C   VISUALIZING CLUSTERS OF NEURAL NETWORKS IN ATTRIBUTES SPACE

In Fig. 4, we clearly see that each network forms a distinct cluster in the attribute space. Also, the networks of the same family, e.g., VGGs, DenseNets, MobileNets, has the clusters close to

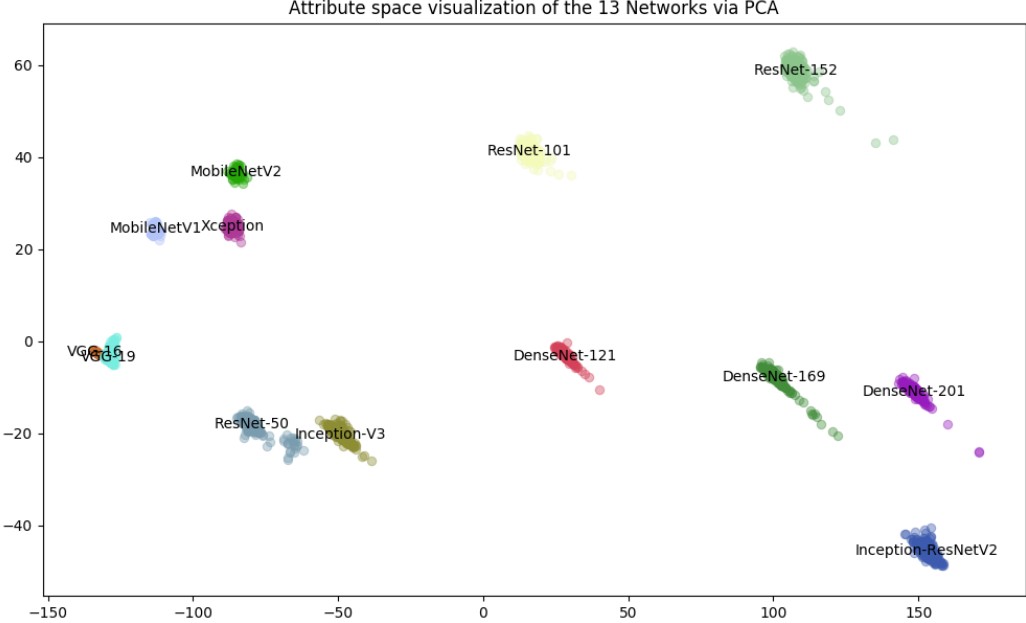

Figure 4: **Attribute space visualization of 13 networks.** To visualize, we perform a principal component analysis (PCA) of the 13 architecture attributes with 650 samples.

|  | *#convs* | *#fcs* | *#softms* | *#relus* | *#mpools* | *#apools* | *#merges* | *#biases* |
|---|---|---|---|---|---|---|---|---|
| **PCA-0** | 0.6715 | -0.0032 | 0.0005 | 0.6294 | -0.0077 | 0.0023 | 0.3891 | -0.0378 |
| **PCA-1** | -0.3857 | -0.0019 | -0.0012 | -0.1383 | -0.0239 | -0.0422 | 0.8599 | -0.3006 |

Table 8: **The scores of 8 attributes in each principal axis.** The first axis (PCA-0) is influenced by *#convs > #relus > #merges*, and the second axis (PCA-1) varies with *#merges > #convs > #relus*.

each other. Table 8 compares the influence scores of each axis. We found that the three attributes discussed in Sec. 4 are the most influential in the two axes.

## D   ATTRIBUTES EXTRACTION RESULTS FROM OTHER NETWORKS

We show the attributes extraction results with the other 11 networks in Table 9.

| Arch. | Data | Attributes | | | | | | | |
|---|---|---|---|---|---|---|---|---|---|
| | | #convs | #fcs | #softms | #relus | #mpools | #apools | #merges | #biases |
| **VGG16** | G | 13 | 3 | 1 | 15 | 5 | 0 | 0 | 16 |
| | S | 13.0 | 3.0 | 1.0 | 15.0 | 4.9 | 0.1 | 0 | 16.0 |
| | L | 13.1 | 3.0 | 1.0 | 15.0 | 4.9 | 0.1 | 0 | 16.0 |
| **ResNet101** | G | 101 | 1 | 1 | 97 | 1 | 1 | 32 | 1 |
| | S | 106.9 | 1.2 | 1.0 | 96.9 | 1.1 | 1.0 | 98.7 | 1.0 |
| | L | 107.2 | 1.2 | 1.0 | 96.9 | 1.1 | 1.0 | 98.7 | 1.0 |
| **ResNet152** | G | 155 | 1 | 1 | 150 | 1 | 1 | 50 | 1 |
| | S | 164.3 | 1.1 | 1.1 | 150.7 | 1.1 | 1.0 | 151.7 | 1.0 |
| | L | 164.3 | 1.1 | 1.1 | 150.7 | 1.1 | 1.0 | 151.7 | 1.0 |
| **DenseNet121** | G | 120 | 1 | 1 | 121 | 1 | 3 | 58 | 1 |
| | S | 125.4 | 1.0 | 1.3 | 119.7 | 1.0 | 3.2 | 59.3 | 1.0 |
| | L | 125.5 | 1.0 | 1.3 | 119.7 | 1.0 | 3.2 | 59.3 | 1.0 |
| **DenseNet169** | G | 168 | 1 | 1 | 169 | 1 | 3 | 82 | 1 |
| | S | 174.6 | 1.0 | 1.2 | 167.8 | 0.9 | 3.3 | 83.0 | 0.9 |
| | L | 175.0 | 1.0 | 1.2 | 167.8 | 0.9 | 3.2 | 83.0 | 1.0 |
| **DenseNet201** | G | 200 | 1 | 1 | 201 | 1 | 3 | 98 | 1 |
| | S | 203.1 | 1.1 | 1.1 | 199.6 | 1 | 3.1 | 99.0 | 1.0 |
| | L | 203.1 | 1.1 | 1.1 | 199.6 | 1.2 | 3.1 | 99.0 | 1.0 |
| **InceptionV3** | G | 94 | 1 | 1 | 76 | 4 | 9 | 15 | 1 |
| | S | 86.9 | 1.1 | 1.0 | 62.9 | 3.9 | 9.0 | 14.7 | 1.1 |
| | L | 86.9 | 1.1 | 1.0 | 62.9 | 3.9 | 9.0 | 14.7 | 1.1 |
| **InceptionResNet** | G | 244 | 1 | 1 | 203 | 4 | 1 | 83 | 40 |
| | S | 234.0 | 1.0 | 0.9 | 186.4 | 4.0 | 1.1 | 84.2 | 40.5 |
| | L | 234.0 | 1.0 | 0.9 | 186.4 | 4.0 | 1.1 | 84.2 | 40.5 |
| **Xception** | G | 41 | 1 | 1 | 36 | 4 | 0 | - | 1 |
| | S | 41.1 | 1.0 | 1.0 | 35.0 | 4.0 | 0 | 41.5 | 1.0 |
| | L | 41.3 | 1.0 | 1.0 | 34.9 | 4.0 | 0 | 41.4 | 1.0 |
| **Mobile (1.0)** | G | 15 | 0 | 0 | 27 | 0 | 0 | - | 1 |
| | S | 15.3 | 0 | 0 | 26.8 | 0 | 0 | 28.3 | 1.0 |
| | L | 15.3 | 0 | 0 | 26.8 | 0 | 0 | 28.4 | 1.0 |
| **MobileV2 (1.0)** | G | 35 | 1 | 1 | 34 | 0 | 0 | - | 1 |
| | S | 35.4 | 1.0 | 1.1 | 34.7 | 0.1 | 0 | 53.3 | 1.0 |
| | L | 35.4 | 1.0 | 1.1 | 34.7 | 0.1 | 0 | 53.2 | 1.0 |

(Note that **G**, **S**, and **L** means **G**round truth, and the observations from **S**hort and **L**ong attacks.)

Table 9: **Extracted architecture attributes of the 11 networks.** We consider the 8 attributes extracted from short and long attacks. In the short attacks, we report the average values observed in ten random queries, whereas in the long attacks, we report the averages of ten continuous queries.

