# OpenReview forum: "Security Analysis of Deep Neural Networks Operating in the Presence of Cache Side-Channel Attacks"
_ICLR.cc/2019/Conference_

### Official Review · AnonReviewer2 · 2018-11-04
**Unclear whether this paper surpasses prior research**

**Rating:** 4
**Confidence:** 2

**Review:**

The paper describes a cache side-channel attack on a deep learning model. In a cache side-channel attack, the attacker sets up a process on the same machine where the victim process (that is running the training or evaluation job for the DNN model) is running. It is assumed that the victim process uses a common shared library for DNN computations as the attacking process. The attacking process flushes the cache, then observes access times for key functions. The paper shows that, based on the speed of accessing previously flushed functions, the attacker can discover the high-level network architecture, namely the types of layers and their sequence. The paper shows that, by spying on such cache access patterns in the Tensorflow library, this method can reliably extract the above high-level information for 11 different network architectures. It also describes a few counterattack alternatives whereby the victim can obfuscate its cache access patterns for self-protection.

The significance of the results is not clear to me. The extracted information is very high level. What realistic attacks can be constructed from such a coarse-grained fingerprinting? The experimental results show that the fingerprint can be used to map the architecture to one of the 13 well-known architectures (VCC16, ResNet, DenseNet, Inception, etc.). But so what? What does the victim lose by revealing that it's using one of a few very well known types of DNNs (the ones tested in this paper). There may very well be a good reason why this is very dangerous, but that is not explained in the paper. Not being familiar with this line of research and its significance, I looked up several of the related papers (Suciu et al., 2018, Tramer et al., 2017, Papernot et al., 2017, Yan et al., 2018). None of them could explain why this particular type of fingerprinting is dangerous.

Of the cited previous work, Yan et al., 2018 seems to present the most closely related approach. The method described in that paper is very similar: cache side attack on a shared library through a co-located attacker process. They monitor at a finer grain -- Generalized Matrix Multiplications -- and are thus able to infer more details such as the size of the layers. This also makes the inference problem harder -- they were able to narrow down the search space of networks from >4x10^35 to 16 (on VGG16). On the surface, the results presented in this paper seem stronger. But they are actually solving a much easier problem -- their search space is one of 13 well-known networks. To me, Yan et al.'s approach is a much more powerful and promising setup.

Overall, while the paper is clearly written and presents the idea succinctly, it is derivative of previous research, and the results are not stronger. I'm not an expert in this area, so it's possible that I missed something. Based on my current understanding, however, I recommend reject.

---

> ### Author Response · Authors · 2018-11-07
> **Clear our contributions over other works (1/2)**
>
> We thank the reviewer for the constructive feedback. We will update the paper accordingly. Additionally, we clarify here the significance of DNN fingerprinting attacks and the relation to the concurrent work of (Yan et al., 2018).
>
> (1) The threat of DNN fingerprinting attacks and the significance of our results.
>
> Prior work on black-box attacks [1, 2, 3] against neural networks assumes an adversary who has knowledge of the victim's network architecture. This is an impractical assumption, and thus, releasing this assumption is the last-mile problem: if an attacker can easily know the architecture of a victim network, this will enable most black-box attacks on DNNs. For instance, without this knowledge, the success of black-box adversarial sample crafting can decrease dramatically, as illustrated in [7]: in attacks against transfer learning services, where the attacker has partial knowledge about the victim network's architecture, having lesser knowledge can decrease the attack's success rate from 88.4% (only the last 3-4/16 layers are unknown) to 1.2% (the last 6/16 layers are unknown). Additionally, DNNs are often proprietary and represent the key intellectual property, and thus their architectures are hidden from attackers. The reconstruction of DNN attributes is also the topic of [4], published at ICLR'18, where the open reviews deemed the problem setting novel and interesting.
>
> What is more, this simple cache side-channel attack is more effective than other network reconstruction attacks proposed in prior work [4, 5]. These approaches are either time intensive (i.e., 40 GPU days for the technique proposed in [4]) or monitor computations while an attacker actively queries a victim model. With our DeepRecon attack, we demonstrate that high-level architectural information --- that prior work aims to extract --- can be easily leaked through our side-channel attacks with little computation and passive monitoring (Sec. 3.2-3.4). This allows an attacker to reconstruct the full network architecture of an arbitrary network (Sec. 3.5) without specifying or assuming knowledge of a network family.
>
> Moreover, our results go beyond proposing and analyzing a fingerprinting attack. We propose a statistical model for fingerprinting to quantify the importance of each piece of leaked information to the attacker's success (Sec. 4). We also propose simple and effective defenses that obfuscate the observations made through cache side-channels, which can be implemented without specific hardware or operating system support (Sec 5).
>
> To the best of our knowledge, this represents the first comprehensive assessment of the vulnerability of DNNs to cache side-channel attacks. We hope that our results will stimulate follow-on work on defending ML systems against such attacks.

---

> > ### Author Response · Authors · 2018-11-26
> > **Clear our contributions over other works (2/2)**
> >
> > (continued: since we are only allowed 5000 chars in each comment.)
> >
> > (2) Comparison to the concurrent work (Yan et al., 2018).
> >
> > The attack proposed in [6] that reconstructs a DNN architecture by monitoring matrix multiplications through cache side-channels has several limitations. First, this attack is highly dependent on the implementation of General Matrix Multiplication (GeMM). There are various libraries such as OpenBLAS, Intel MKL, Atlas, or NVBLAS that implement GeMM differently, thus, the attacker in [6] needs to need to spend a lot of time reverse engineering the framework they are attacking (Sec 4, 5, 6 in [6]). Also, the GeMM-based reverse engineering can only reveal convolutional or fully connected layers because others such as activations and pooling layers are difficult to characterize by matrix computations. They realize and state this limitation in their paper (Sec 4.4), and probe the addresses of functions that implement these layers, e.g., relu, softmax, or tanh. Our attack takes this simpler approach and attacks control flow functions in the core of machine learning frameworks, suggesting that our approach is more general and applies in more settings. Moreover, the attack in [6] assumes the victim is using the OpenBLAS library, a library built for CPUs. We stress that this approach of attacking code in the CPU implementation of GeMM does not generalize to attacks on a GPU. To monitor GeMM computations running on a GPU, the GPU's cache needs to be shared between the colocated attacker process and the victim's process. This means an attack against a victim's network model on a GPU using this approach would be ineffective. On the other hand, since we attack the general functions, which is implemented and run on the CPU regardless of where the matrix multiplication takes place, we make a stronger claim in (Sec. 3.3) that DeepRecon is independent of hardware --- i.e., we can target a DNN model running on CPUs or GPUs.
> >
> > Additionally, DeepRecon does not assume knowledge of the victim's network family unlike the attack proposed in [6]. The attack in [6] is only able to reduce search space within a network family, e.g., in their experiments (Sec. 7), they find several candidate network architectures within one specific network family (i.e., VGG or ResNet). In contrast, our attacker can reduce the search space dramatically by using leaked information to classify an arbitrary victim's network into a network family. We can then use this information to simplify the reconstruction of the network architecture fully (Sec 3.4-3.5). We also show that DeepRecon can be applied in transfer learning cases to identify pre-trained networks inside a victim's DNN by identifying the parent model and the layer at which the backpropagation is frozen (Sec 3.3).
> >
> > An even more important distinction from [6] is that we implement and evaluate two defenses against fingerprinting attacks that exploit cache side channels. As our defenses take advantage of standard DNN computations and do not assume a particular attack implementation, they should be effective against the attacks proposed in [6].
> >
> > Finally, we note that [6] represents concurrent and unpublished work. The paper does not appear to have been published yet in a conference or journal, and it was uploaded to ArXiv on Aug. 14, 2018 (44 days before the ICLR'19 deadline). As our research was conducted primarily in June and July 2018, it is not a derivative of [6] but rather an independent and concurrent project.
> >
> >
> > [1] Suciu, Octavian, et al. "When Does Machine Learning FAIL? Generalized Transferability for Evasion and Poisoning Attacks." arXiv preprint arXiv:1803.06975 (2018).
> > [2] Tramèr, Florian, et al. "Stealing Machine Learning Models via Prediction APIs." USENIX Security Symposium. 2016.
> > [3] Papernot, Nicolas, et al. "Practical black-box attacks against machine learning." Proceedings of the 2017 ACM on Asia Conference on Computer and Communications Security. ACM, 2017.
> > [4] Oh, Seong Joon, et al. "Towards reverse-engineering black-box neural networks." ICLR. 2018.
> > [5] Hua, Weizhe, Zhiru Zhang, and G. Edward Suh. "Reverse engineering convolutional neural networks through side-channel information leaks." Proceedings of the 55th Annual Design Automation Conference. ACM, 2018.
> > [6] Yan, Mengjia, Christopher Fletcher, and Josep Torrellas. "Cache telepathy: Leveraging shared resource attacks to learn DNN architectures." arXiv preprint arXiv:1808.04761 (2018).
> > [7] Bolun Wang, Yuanshun Yao, Bimal Viswanath, Haitao Zheng, and Ben Y Zhao. With great training comes great vulnerability: Practical attacks against transfer learning. USENIX Security, 2018

---

### Official Review · AnonReviewer3 · 2018-11-08
**Simple yet effective attacks to infer model architectures; more clarification would help**

**Rating:** 6
**Confidence:** 4

**Review:**

This paper performs cache side-channel attacks to extract attributes of a victim model, and infer its architecture accordingly. In their threat model, the attacker could launch a co-located process on the same host machine, and use the same DL framework as the victim model. Their evaluation shows that: (1) their attacks can extract the model attributes pretty well, including the number of different types of layers; (2) using these attributes, they train a decision tree classifier among 13 CNN architectures, and show that they can achieve a nearly perfect classification accuracy. They also evaluate some defense strategies against their attacks.

Model extraction attack under a black-box setting is an important topic, and I am convinced that their threat model is a good step towards real-world attacks. As for the novelty, although Yan et al. also evaluate cache side-channel attacks, that paper was released pretty shortly before ICLR deadline, thus I would consider this work as an independent contribution at its submission.

I have several questions and comments about this paper:

- One difference of the evaluation setup between this paper and Yan et al. is that in Yan et al., they are trying to infer more detailed hyper-parameters of the architecture (e.g., the number of neurons, the dimensions of each layer, the connections), but within a family of architectures (i.e., VGG or ResNet). On the other hand, in this paper, the authors extract higher-level attributes such as the number of different layers and activation functions, and predict the model family (from 5 options) or the concrete model architecture (from 13 options). While I think inferring the model family type is also an interesting problem, this setup is still a little contrived. Would the classifier predict the family of a model correctly if it is not included in the training set, say, could it predict ResNet32 as R (ResNet)?

- In Table 3, it looks like the errors in the captured computation sequences show some patterns. Are these error types consistent across different runs? Could you provide some explanation of these errors?

- In Table 5, my understanding is that we need to compare the avg errors to the numbers in Table 2. In this case, the errors seem to be even larger than the sum of the attribute values. Is this observation correct? If so, could you discuss what attributes are most wrongly captured, and show some examples?

- It would be beneficial to provide a more detailed comparison between this work and Yan et al., e.g., whether the technique proposed in this work could be also extended to infer more fine-grained attributes of a model, and go beyond a classification among a pre-defined set of architectures.

- The paper needs some editing to fix some typos. For example, in Table 5, the captions of Time (Baseline) and Time (+TinyNet) should be changed, and it looks confusing at the first glance.

---

> ### Author Response · Authors · 2018-11-13
> **More clarifications**
>
> We thank the reviewer for the constructive feedback: the questions and comments can improve and make our contributions more concrete. We will update our paper accordingly to include their points. In the meantime, we would like to provide initial answers to the reviewer’s questions:
>
> (1) Could our classifier predict the family of a model correctly (ex. ResNet32) not in the training data?
>
> No, our classifier could not predict this because it cannot learn how to classify unobserved samples into the families that have similar features (or attributes). Suppose that the samples from ResNet18 or 34 are not in our training set. Since the architecture attributes of ResNet18 or 34 are similar to those of VGG16/19 or MobileNetV1/2, our classifier may predict the unseen samples as some other close family (VGGs or MobileNets). However, we are sure that if we include the ResNet18 or 34 to our training set, our classifier will learn to specify them as ResNets.
>
> The key contribution of our (fingerprinting) experiment is to examine which of the architecture attributes that our attacker can extract are essential to specify network families. We identified that four common attributes (#relus, #merges, #convs, and #poolings) are important to know the family of a victim’s network. This information can help our attacker to launch large-scale attacks in the transfer learning scenario because our attacker already knows multiple commonly used pre-trained models + architectures that she can train her classifier on. Then, by passively observing the information leakage from cache side-channels, the attacker can specify which actual pre-trained model that the victim uses and synthesize adversarial samples with the pre-trained model that also works for the victim model (as prior work [1] warned).
>
> (2) Reconstruction errors observable in Table 3.
>
> In our experiments, we could not find specific error patterns in the extracted attribute sequences. As we can see in Table 3, there are the cases where convolutional layers are missing and/or added and activations are missing and/or added. Also, the locations of missing attributes are different in each run. We attribute these errors to a few primary causes: there is background noise of other processes that our flush+reload cache-based side channel attack may pick up (e.g. other background processes pull something into the cache and evict our target functions between when the victim calls the function and we reload it), or we may experience common errors associated with flush+reload (e.g. a victim may call the function during the time when we reload, causing us to see a cache miss instead of correctly observing a cache hit) [2].
>
> (3) Comparison of avg. errors in Table 5 (running decoy process as a defense).
>
> Yes, in Table 5, our experiments indicate that the errors are larger than the sum of the original attribute values (that we can expect from ResNet50). In our experiments in Table 5, we increase the errors associated with the attributes that we aim to obfuscate. For instance, when we run the TinyNet with only one convolutional layer, we observe the #conv attribute is significantly increased. This result is important because, with our defenses, a defender can choose the attributes to obfuscate. By introducing noise into the cache side channel by means of another process, we can make differentiating between functions that are called by our victim and our decoy incredibly difficult and therefore mitigate, and possibly eliminate, any useful information that an attacker can gain by these side channels. Since the defender has control over what noise gets introduced, they can also dynamically and adaptively change what noise is added into the attacker’s observations, thereby increasing our defenses’ effectiveness and generalizability.
>
> (4) Emphasizing our contributions over the concurrent work (Yan et al., 2018).
>
> Our key contributions over the concurrent work (Yan et al., 2018) are highlighted in the initial response to the reviewer’s comments below [comment 1]: https://openreview.net/forum?id=rk4Wf30qKQ&noteId=B1l2z9wgTm / comment 2: https://openreview.net/forum?id=rk4Wf30qKQ&noteId=Sye8V5wx67]. We plan to include the comparison in our related work section.
>
> (5) Fixing typos in our paper.
>
> We are working on revising our paper based on the reviewers’ feedback. We will include those fixes in the revised version.
>
> [1] Bolun Wang, Yuanshun Yao, Bimal Viswanath, Haitao Zheng, and Ben Y Zhao. With great training comes great vulnerability: Practical attacks against transfer learning. USENIX Security, 2018
> [2] Yarom, Yuval, and Katrina Falkner. "FLUSH+ RELOAD: A High Resolution, Low Noise, L3 Cache Side-Channel Attack." USENIX Security Symposium. Vol. 1. 2014.

---

### Official Review · AnonReviewer1 · 2018-11-08
**Unclear threat model with a very strong adversary that obtains information of moderate significance.**

**Rating:** 4
**Confidence:** 4

**Review:**

This paper considers the problem of fingerprinting neural network architectures using cache side channels. In the considered threat model, the attacker runs a process co-located with the victim's, and uses standard FLUSH+RELOAD attacks to infer high-level architectural information such as the number and types of layers of the victim's ML model. The paper concludes with the discussion of some "security-through-obscurity" defenses.

I don't quite understand the threat model considered in this paper. The main motivating factor given by the authors for uncovering model architecture details is for facilitating black-box attacks against ML models (e.g., for adversarial examples or membership inference).
Yet, in the case of adversarial examples for instance, knowledge of the architecture is often considered a given as keeping it secret has very little influence on attacks. There are black-box attacks that require no knowledge of the architecture and only a few queries (e.g., Black-box Adversarial Attacks with Limited Queries and Information, Ilyas et al., ICML'18).
So overall, learning such coarse-grained features about a model just doesn't seem particularly useful, especially since architecture-level details are often not considered private or secret to begin with.

After architectural details have been extracted, the end-goal attacks on ML models considered by the authors (e.g., model stealing, adversarial examples, etc.) require query access anyways. Thus, additionally assuming co-location between the adversary and the victim's model seems to unnecessarily strengthen the attacker model.

Maybe the most interesting scenario to consider for cache side-channels in ML is when ML models are run on trusted hardware (e.g., Oblivious Multi-Party Machine Learning on Trusted Processors, Ohrimenko et al.; or this work also submitted to ICLR: https://openreview.net/forum?id=rJVorjCcKQ).
Cache side channels are much more relevant to that threat model (i.e., ML code running in a trusted hardware enclave hosted by a malicious party). And indeed, there have been many cache side-channel attack papers against trusted hardware such as Intel's SGX (e.g., Software Grand Exposure: SGX Cache Attacks Are Practical, Brasser et al.)

But given what we know about the strength of these cache side channel attacks, one would expect to be able to extract much more interesting information about a target model, such as its weights, inputs or outputs. In the above trusted hardware scenario, solely extracting architecture-level information would also not be considered a very strong attack, especially since coarse-grained information (e.g., a rough bound on the number of layers), can be trivially obtained via timing side channels.

Minor comments:
- In the introduction, you say that white-box attacks for adversarial examples are rendered ineffective by gradient masking. This isn't true in general. Only "weak" white-box attacks can be rendered ineffective this way. So far, there are no examples of models that resist white-box attacks yet are vulnerable to black-box attacks.
- What exactly causes the cache-level differences you observe? Can you give some  code examples in the paper that showcase what happens? Are the TensorFlow code lines listed in Table 1 from a specific commit or release?
- The defenses discussed in Section 5 are all forms of "security through obscurity" that seem easily defeated by a determined attacker that adapts its attack (and maybe uses a few additional observations).

--REVISION--
I thank the authors for their rebuttal and clarifications on the threat model and end goals of their attacks. I remain somewhat unconvinced by the usefulness of extracting architectural information. For most of the listed attacks (e.g., building substitute models for adversarial examples, or simply for model extraction) it is not clear from prior work that knowledge of the architecture is really necessary, although it is of course always helpful to have this knowledge. As I mentioned in my review, with current (undefended) ML libraries, it should be possible to extract much more information (e.g., layer weights) using cache side channels.

---

> ### Author Response · Authors · 2018-11-14
> **Clarification of our threat model and state our contributions out clearly (1/2)**
>
> We thank the reviewer for the constructive feedback. We will update the paper to convey our contributions clearly. Here, we provide the clarification of our threat model and state our key contributions over other works.
>
> (1) Clarification of our threat model.
>
> The two initial concerns are “the arch. information that we extracted does not seem useful” and “the network architecture details are often not considered private or secret.” However, this is often not the case. First, our attack enables a continuous threat since the extracted information helps our attacker easily launch various attacks, such as black-box evasion attacks [3], model extractions, or adversarial attacks in transfer learning scenarios [6], with minimal overhead. For instance, to cause multiple evasions, the black-box attack in [3] needs to repeat the attack for multiple targets since it only approximates the gradients around a single target instance. On the other hand, our attack allows the attacker to estimate a surrogate model and lets her easily synthesize multiple evasion samples.
>
> We also address the reviewer’s concern that “our attacker requires query accesses to the victim model anyway after reconstructing network architectures of victims.” However, this is not necessarily true. Suppose that our attacker has a similar dataset or a part of the victim’s training dataset --- this assumption is reasonable because many datasets for a common task such as face recognition are openly and freely available in online. Then, the attacker separately trains the reconstructed network offline with her own data and uses it as a surrogate model, which does not require query accesses to the model after the reconstruction.
>
> Because of the reasons above, prior work has considered various attackers who aim to extract the network architecture details [1, 4, 5] as well as defenses that keep this information secret, as shown in paper [2] mentioned by the reviewer (i.e., “Model privacy w.r.t. The server and client” defined in Sec 2.1). These contributions only make sense in a setting where the architecture is secret.
>
> The last issue raised is that both the co-location and query access assumptions make our adversary stronger than the black-box attacker who only uses query accesses. However, our attacker does not actively query a victim model, which is a non-trivial difference from the black-box attacker. Our attacker passively monitors the model’s computations and utilizes the information leakages. Suppose an attacker aims to install malware on a machine where an anti-virus system, based on a DNN model, is running. In the black-box attacks, the attacker actively drops files to monitor the model’s decisions. On the other hand, our attacker has the victim install a chrome add-on that monitors cache behaviors to extract the arch. information of the model. In this case, our attacker is less likely to be detected. Thus, we claim that compared to the black-box attackers, there are cases in which our attack does not assume a stronger adversary. We, therefore, conclude that the settings in which these attacks might be useful to an attacker are often separate and distinct.
>
> (2) Comparison with trusted hardware cases.
>
> Yes, (cache) side-channel attacks against DNNs running on trusted hardware can be interesting direction since concurrent work has proposed hiding computation details of a DNN with this type of hardware. However, our work delivers separate, non-trivial contributions from [2]. We note that the kind of information that can be extracted from DNN models through cache side-channel attacks is an open question. Thus, our work provides an answer: an attacker can extract the same level of information in current deep learning setups via side-channels that is considered to be important to perform further sophisticated attacks in prior work [1, 4, 5, 6], without queries or extensive computations. Also, cache side-channel and timing side-channel attacks often only work when victim’s computations are data-dependent and are performed on specific hardware (e.g. on a victim running AES128 encryptions on a CPU), whereas we show that cache attacks on DNNs can extract the targeted information regardless of what data is being processed and on what hardware it is being performed (e.g., on any prediction input and on CPUs or GPUs).

---

> > ### Author Response · Authors · 2018-11-26
> > **Clarification of our threat model and state our contributions out clearly (2/2)**
> >
> > (3) About minor comments.
> >
> > (3) - 1) The comments about the introduction: Yes, we agree with the reviewer that all white-box attacks cannot be defended by gradient masking. Hence, we will fix this claim in the revised version. What we will explain instead is that complete white-box assumptions are not practical, and attackers can spare time and effort when we lessen the black-box assumptions through side-channels.
> >
> > (3) - 2) Our attacker keeps monitoring the time to access the lines of code (or instructions) identified in the shared deep learning framework. If a victim uses specific functions while processing data with a model, the lines in the functions are in the instruction cache. Hence, the access time observed from our attacker will be short. On the other hand, if the code is not in the instruction cache, the access time to the same code will be much longer. From this difference, the attacker identifies which functions are used. We utilize Flush+Reload attack implemented in Mastik toolkit [https://cs.adelaide.edu.au/~yval/Mastik/]. Also, we extracted the lines of code from TensorFlow v1.9.0-rc0.
> >
> > (3) - 3) In the decoy processes, our defenses are difficult to defeat, even by an advanced attacker. This is because the attacker cannot separate the information observed into separate processes and therefore models. Once the DL framework function call is loaded into the instruction cache, the instruction is cached in the same manner between both the victim and decoy processes. This limits the attacker from observing which accesses are from which process and therefore mitigates, and possibly eliminates any useful information an attacker may hope to observe. Also, even if an attacker finds a method to differentiate the information into separate processes, a defender can dynamically and adaptively inject different obfuscating data into the attacker’s observations by running a different model. In the oblivious model computations, we can randomize the computation orders of unraveled paths while each query is being processed.
> >
> > Thus, we emphasize that these defenses do not represent “security through obscurity”. Because cache side-channels only provide to an attacker coarse-grained timing information from all other processes colocated on the machine, adding a significant amount of noise by means of another process and model architecture is sufficient to make extracting meaningful data incredibly difficult—even when the adversary knows all the details of the defense (including our implementation).
> >
> > [1] Oh, Seong Joon, et al. "Towards reverse-engineering black-box neural networks." ICLR. 2018.
> > [2] Tramer, Florian, and Dan Boneh. "Slalom: Fast, Verifiable and Private Execution of Neural Networks in Trusted Hardware." arXiv preprint arXiv:1806.03287 (2018).
> > [3] Ilyas, Andrew, et al. "Black-box Adversarial Attacks with Limited Queries and Information." arXiv preprint arXiv:1804.08598 (2018).
> > [4] Hua, Weizhe, Zhiru Zhang, and G. Edward Suh. "Reverse engineering convolutional neural networks through side-channel information leaks." Proceedings of the 55th Annual Design Automation Conference. ACM, 2018.
> > [5] Yan, Mengjia, Christopher Fletcher, and Josep Torrellas. "Cache telepathy: Leveraging shared resource attacks to learn DNN architectures." arXiv preprint arXiv:1808.04761 (2018).
> > [6]  Bolun Wang, Yuanshun Yao, Bimal Viswanath, Haitao Zheng, and Ben Y Zhao. With great training comes great vulnerability: Practical attacks against transfer learning. USENIX Security, 2018

---

### Author Response · Authors · 2018-11-26
**Paper Revision**

We thank our reviewers for taking the time to read, evaluate our work, and provide constructive feedback. We have uploaded a revised version of our paper, with edits to address the concerns raised. We summarize our updates below:

1. We update the content that made our contributions confusing in Sec. 1.
   (e.g., black-box attacks -> attacks on black-box models, etc.)

2. We further discuss the concurrent work (Yan et al.) in Sec. 2.

3. We provide an example attack scenario in Sec. 3.1.

4. We provide limitations to our approach and hypothesize the effect of these results on future work in Sec. 3.2.

5. We provide an analysis of the reconstructions errors observed in Table 3 in Sec. 3.5.

6. We provide some more discussion about what our attack can do in the transfer learning setting after fingerprinting the victim network in Sec. 4.

7. We update Table 5. to show the individual attribute errors in Sec. 5.1

8. We emphasize that our defender can adaptively choose the attributes to obfuscate by decoy processes in Sec. 5.1.

9. We fix typos and improve writing throughout the entire paper.

Please see our replies to each reviewer for our detailed responses to individual points.

---

### Meta-Review · Area_Chair1 · 2018-12-16
**results too weak given strength of attacker**

**Confidence:** 4
**Recommendation:** Reject

**Metareview:**

The reviewers generally had concerns that the goal of recovering only the model architecture was unmotivated (given that knowing the architecture is not a large threat on its own, and there are existing attacks that work without knowledge of the model architecture). Moreover, given the strength of the assumed attack model, recovering model architecture is a fairly unambitious goal (again, more serious attacks have already been demonstrated under weaker attack models). Finally, though less seriously, the analysis is fairly preliminary, e.g. it is unclear if the attack can generalize to nearby architectures that were outside the training set.